# Study of Mechanical and Thermal Properties of Environmentally Friendly Composites from Beer Bagasse

**DOI:** 10.3390/polym16202916

**Published:** 2024-10-17

**Authors:** María Jordá-Reolid, Asunción Martínez-García, Ana Ibáñez-García, Miguel Ángel León-Cabezas, Josefa Galvañ-Gisbert

**Affiliations:** Innovative Materials and Manufacturing Area, Technological Institute for Children’s Products and Leisure, 03440 Ibi, Spain; sunymartinez@aiju.es (A.M.-G.); anaibanyez@aiju.es (A.I.-G.); miguelangelleon@aiju.es (M.Á.L.-C.); pepigalvany@aiju.es (J.G.-G.)

**Keywords:** biocomposites, beer bagasse, BioPE, PLA, PP, circular economy, sustainability, compatibilisation

## Abstract

The influence of bagasse fibres from beer manufacturing in mechanical, thermal, and rheological properties of three polymers (BioPE, PLA, and PP) has been studied in order to develop new environmentally friendly biocomposites for injection moulding applications. Totals of 10 wt%, 20 wt%, and 30 wt% of bagasse fibre (BSG) were added to the polymers by extrusion compounding, adding specific compatibilising additives, and injected samples were mechanically characterised by tensile, Charpy impact, and hardness tests. In addition, the fractures obtained after the impact test were observed using scanning electron microscopy (SEM) to assess the compatibility matrix filler. Characterisation of the thermal properties is also carried out by using differential scanning calorimetry (DSC) and thermogravimetry (TGA). Additionally, melt flow index of the biocomposites is also studied. An increase in the rigidity of the BioPE and PP composites was produced with the increase in BSG content, dealing with a decrease in maximum strain and impact resistance; whereas, in the filled BGS PLA biocomposites, Young’s modulus was lower than that of the PLA material, improving the ductility of the PLA-BGS formulations. Compatibilisation effect was, therefore, different in the nine developed formulations, and the BGS content also influenced their thermal, mechanical, and rheological behaviours.

## 1. Introduction

The environmental impact of plastics has emerged as a significant global concern, particularly in recent decades, as the accumulation of thermoplastic waste in natural ecosystems has reached critical levels. Traditional plastics, primarily derived from non-renewable petrochemical sources, are resistant to degradation, leading to persistent environmental pollution and adverse effects on wildlife and marine ecosystems [1]. Additionally, the production, usage, and disposal of these plastics contribute significantly to greenhouse gas emissions, exacerbating the global climate crisis [2,3].

In response to these pressing issues, research has increasingly focused on developing sustainable alternatives to conventional plastics. Biodegradable polymers, such as polylactic acid (PLA) and polyhydroxyalkanoates (PHAs), offer a promising solution due to their ability to decompose under specific environmental conditions, thereby reducing their long-term ecological impact [4]. Additionally, recent advancements have explored the integration of renewable resources into the production of bioplastics. This includes the use of agricultural by-products and bio-based feedstocks, which not only diminish dependence on fossil fuels but also align with circular economy principles, promoting resource efficiency and waste minimisation [4]. These innovations represent a pivotal shift towards more sustainable practices in plastic production and usage, aiming to mitigate the environmental challenges posed by traditional plastics.

The escalating environmental concerns related to traditional plastics have spurred significant interest in sustainable alternatives, with Wood Plastic Composites (WPCs) emerging as a promising option. WPCs are sustainable materials combining plant fibres and thermoplastics, offering improved properties over their individual components [5]. Natural fibres from agro-food waste, such as jute, hemp, olive branches, almond shells, and argan shells, are increasingly used as reinforcements in WPCs [6,7,8]. These composites exhibit enhanced mechanical properties, thermal stability, and fire resistance compared to neat polymers [7,8,9]. However, challenges include poor compatibility between hydrophilic fibres and hydrophobic matrices and needing surface treatments or compatibiliser additives to improve adhesion [10,11]. WPCs can be manufactured using various methods, including extrusion, injection moulding, and advanced techniques like powder bed fusion additive manufacturing [11]. These eco-friendly composites have potential applications in automotive, aerospace, and construction industries, offering a promising alternative to conventional materials [10,11,12].

Recent research has explored the potential of various nut shells and agricultural wastes as fillers in biocomposites. Almond shell powder has been incorporated into polyester-based matrices, improving elastic modulus and thermal stability [13]. Pecan nutshell powder showed good dispersion and adhesion in polylactic acid (PLA) biocomposites [14]. Thermoplastic starch biocomposites with nut shells demonstrated improved strength parameters [15]. Almond shell varieties in starch-based biocomposites showed no significant differences in mechanical properties [16]. Argan nutshell reinforced PLA biocomposites exhibited improved mechanical properties with surface treatments [17]. Polypropylene biocomposites with argan nutshell showed enhanced properties with coupling agents [18]. These studies highlight the potential of nut shells and agricultural wastes in developing sustainable, biodegradable materials for various applications, including packaging and disposable items.

Another fibre used in the manufacture of WPC biocomposites is BSG. Bagasse, a residue from beer production, is rich in fibres and proteins, making it a valuable resource for various applications [19]. It can be used to create fibre-enriched food products like honey cakes and bread [20,21] as well as bioplastics and biocomposites for 3D printing [22,23]. Bagasse fibres have been studied for use in nonwoven materials [24] and as additives in asphalt mixtures, showing improved performance comparable to lignin fibres. Using bagasse in these applications not only adds value to the waste product but also contributes to environmental protection and sustainability in various industries. In addition, BSG represents about 85% of the by-products generated in the beer production process, which corresponds to 31% of the weight of the original malt. The proportion of BSG obtained is 20 kg per 100 litres of beer produced [25], so large volumes are obtained at low cost throughout the year. Annual world production is estimated at 30 million tons [26] and in the European Union approximately 3.4 million tons [27]. This would allow between 3′74 and 4′42 bioplastic production. Interest from industrial sectors has been received in the research centre where the authors develop these composites, with the aim of incorporating bio-based formulations in their products. As a consequence, research has recently started within the framework of a European project [28] to develop composites with BGS fibres to apply and validate them in industrial sectors (furniture and pallets), involving a step of BSG fibre preparation, requirements, and specification of the bio-based new formulations and development of injection moulded demonstrators. These formulations can be extended to other types of consumer products, such as toys, childcare, packaging, or household, where the importance of reducing consumption of non-renewable raw materials is increasingly greater.

In this article, three percentages of beer bagasse fibres (10 wt%, 20 wt%, and 30 wt%) were added to three different polymeric matrices: polyethylene from renewable sources (BioPE), biodegradable polylactic acid (PLA), and polypropylene (PP). The effects of the addition of BSG on the mechanical, thermal, and rheological properties were studied. Specific compatibiliser or coupling agent additives have been added to each formulation to improve bonding of polymeric matrix to the incorporated bagasse fibres. These types of additives provide, in general, higher tensile and flexural strength, increased stiffness at high strains, and sometimes improved toughness by increasing the interfacial stress required to debond the filler–polymer interface [29].

## 2. Materials and Methods

### 2.1. Materials

In this study, three different polymers are used as a polymer matrix to blend with beer bagasse fibres as a filler to study the effect in different technical characteristics of using three different percentages (10 wt%, 20 wt%, and 30 wt%) of filler. To make the polymer matrix compatible with the filler, specific compatibiliser additives for these polymers are used.

Polymers studied are a bio-based polyethylene (BioPE), a polylactic acid (PLA), a bio-based and biodegradable polymer, and a polypropylene (PP).

The bio-based polyethylene (BioPE) used is Terralene HD4527 supplied by FKuR Kunststoff GmbH (Willich, Germany), with a melt flow index (MFI) between 13 and 15 g/10 min (190/2.16) and a modulus of elasticity of 1745 MPa. The PLA used is Beograde INJ038 supplied by Beologic (Zwevegem, Belgium), with an MFI of 17 g/10 min (190/2.16) and a modulus of elasticity of 1610 MPa. The bioPP reference studied is ISPLEN^®^ PB 150G2M from Repsol YPF S.A. (Madrid, Spain), with MFI of 7 g/10 min (230/2.16) and a flexural modulus of 1150 MPa.

The beer bagasse (BSG) is supplied in the form of dry powder by L. Pernía (Albolloque, Guadalajara, Spain) with a particle size between 180 μm and 2 mm. Figure 1 shows the TGA curve of BSG, where weight loss is observed. The first weight loss between 30 °C and 150 °C corresponds to the loss of moisture in the fibres, the second between 150 °C and 400 °C, which corresponds to the degradation of BSG cellulose, a third between 400 °C and 600 °C, which corresponds to lignin, and finally, a fourth that corresponds to the degrasing of the oxidisable groups of BSG, as it coincides with the input of O_2_ on the TGA test [30]. Figure 2 shows the FTIR spectrum of BSG. In this case, characteristic peaks of different bonds in the chemical structure of BSG are observed: peaks located in 1750–1700 cm^−1^ region, corresponding to the C=O bond. This peak is associated with the hemicellulose present in the bagasse. The samples also present carboxyl and amino bonds (1800–1000 cm^−1^), which correspond to proteins and polysaccharides [22].

Three commercial compatibiliser additives are selected according to their suitability for each of the studied polymeric matrices with lignocellulosic fibres. Specifically, the POLYBOND™ 3009 additive from SI Group (The Woodlands, TX, USA) based on polyethylene with maleic anhydride graft (PE-g-AM) is used for BioPE biocomposites; the XIBOND™ 250 additive from Polyscope (Indianapolis, IN, USA) based on random copolymers of styrene and maleic anhydride is used for PLA biocomposites; and for the PP, the additive POLYBOND™ 3200 from SI Group (The Woodlands, TX, USA), which is a chemically modified PP, is used.

According to the supplier, POLYBOND™ 3200 and POLYBOND™ 3009 are recommended modifier additives specially designed for improving chemical coupling for cellulose fibre type fillers (among others) for PP [31] and PE [32] composites, respectively. They can enhance physical, mechanical, and thermal properties even with low addition levels as well as improved processability.

In a study comparing different types of maleic anhydride-grafted polypropylene (MAPP) on the interfacial adhesion properties of bio-flour-filled polypropylene composites [32], the authors conclude that this specific additive has higher MA graft (%) content, MFI with low polydispersity, and higher thermal stability and decomposition temperature. These properties help the improvement of mechanical properties of the POLYBOND modified PP composites compared to those of non-additivated composites due to the enhanced interfacial adhesion of the MAPP composites [29].

Other authors studied the effects of POLYBOND™ 3009 and other coupling agent’s based on acrylated functional monomer on the mechanical properties of HDPE–wood–flour composites [33], concluding that the coupling agents significantly enhanced flexural and tensile strengths, with the maleated polyolefin ones performing better.

Regarding the additive, XIBOND™ 250 is designed as a viscosity and surface modifier, compatibiliser, adhesion promoter, extender, and coupling agent [34]. It improves tensile properties as a coupling agent, providing low viscosity and improved creep and tensile properties. These XIBOND additives have been used in some research to improve plastification and compatibilisation in composites based on PLA and organic fillers (i.e., lignocellulosic waste from chia seed [35]) or to enhance toughness of PLA/PC (80/20 wt%) blends [35].

### 2.2. Biocomposite Preparation

Biocomposites based on three materials are prepared by adding bagasse fibres in three percentages: 10 wt%, 20 wt%, and 30 wt% to BioPE, PLA, and PP. A total of 3 phr (per hundred resin) of corresponding compatibilising is added to all biocomposites to improve the interaction between the bagasse fibres and the polymer. Table 1 includes a list of the biocomposites developed and the nomenclature given in the study.

#### 2.2.1. Drying

For correct processing of the biocomposites, the bagasse is dried before extrusion compounding at 80 °C for 6 h until the moisture is less than 1 wt% by using a UN30 oven from Memmert GmbH (Büchenbach, Germany). Biocomposites obtained by extrusion compounding, as explained in Section 2.2.2, are also dried at 80 °C for 6 h before injection moulding.

#### 2.2.2. Extrusion Compounding Process

The extrusion compounding for the preparation of the biocomposites is carried out with the extrusion line formed by the twin-screw extruder, Collin Teach Line ZK25, the Collin Teach Line WB850 bath, and the Collin Teach Line SP cutter from COLLIN Lab & Pilot Solutions GmbH (Maitenbeth, Germany). A total of 3 kg of each formulation is prepared. Table 2 shows the extrusion compounding conditions programmed for each group of biocomposites.

Figure 3 includes the extrusion compounding equipment used to develop the biocomposites (a) as well as some examples of the obtained biocomposites of BioPE (b), PLA (c), and PP (d) in the form of pellets.

#### 2.2.3. Injection Moulding Process

The injection moulding of specimens is carried out in a Demag Ergotech 7174-0041 injection moulding machine from Demag Plastics Machinery Gm (Schwaig, Germany). Standardised 1A specimens are injected (Figure 4) with the injection moulding conditions shown in Table 3.

### 2.3. Biocomposite Characterisation

The biocomposites included in Table 1 are characterised by thermal, rheological, mechanical, and morphological analyses.

#### 2.3.1. Thermal Characterisation

The degradation onset temperatures (T_onset_) and maximum speed of degradation temperature (T_dmax_) of the developed biocomposites are determined by thermogravimetric analysis (TGA), melting temperature (T_m_), and glass transition temperature (T_g_) by means of differential scanning calorimetry (DSC) tests carried out. Three repetitions of each test are performed, obtaining the mean value. The DSC test is carried out by DSC Q200 equipment from TA Instruments (New Castle, DE, USA), using a heating–cooling–heating cycle: from T_amb_ to 200 °C, from 200 °C to −90 °C, and from 90 °C to 250 °C, with an N_2_ flow rate of 50 mL/min. TGA is carried out with TGA Q500 equipment from TA Instruments (New Castle, DE, USA), using a heating ramp from T_amb_ to 600 °C with an atmosphere of N_2_ with a flow of 50 mL/min and a second ramp from 600 °C to 1000 °C with an O_2_ flow of 50 mL/min.

#### 2.3.2. Rheological Characterisation

The melt flow index (MFI) of the biocomposites was carried out according to the UNE EN ISO 1133-1 standard [36] with Atsfaar’s Twelvindex equipment from ASTFAAR (Milan, Italy) at a test temperature of 190 °C and a mass of 2.16 kg. Ten repetitions of each test are performed, obtaining the mean value.

#### 2.3.3. Mechanical Characterisation

The mechanical characterisation tests performed are tensile, flexural, Charpy impact, and hardness. Instron 6025 Universal Testing Machine equipment from Instron (Norwood, MA, USA) are used for tensile tests, with a 10 kN load cell. The tensile tests are carried out according to the UNE EN ISO 527 standard [37], using type 1A dog bone specimen. The speed for the determination of the modulus is 1 mm/min, and for the rest of the test, it is 500 mm/min. Five repetitions of each test are performed, obtaining the mean value. The impact tests are carried out with Ceast Resil Impactor equipment with a 1 J pendulum according to the UNE EN ISO 179 standard [38], and specimens of 80 mm x 10 mm × 4 mm are used. Ten repetitions of each test are performed, obtaining the mean value. The hardness is determined according to the UNE-EN ISO 868 standard [39] using Bareiss BS-61 hardness equipment from Bareiss Prüfgerätebau GmbH (Oberdischingen, Germany). Five repetitions of each test are performed, obtaining the mean value.

#### 2.3.4. Morphological Characterisation

Scanning electron microscopy (SEM) is used to study the morphology of the biocomposites of the fractures obtained after the Charpy impact test is performed with the microscope IT500HR/LA from Jeol (Tokyo, Japan) at a voltage of 5.0 kV. 

## 3. Results and Discussion

The DSC curves of the BioPE, PLA, and PP formulations are shown in Figure 5, Figure 6 and Figure 7.

Figure 5 shows a single peak, which corresponds to the melting point of BioPE, which is found between 131 °C and 133 °C in the different formulations based on BioPE. This peak becomes smaller as the amount of BSG increases. This is because when BSG is incorporated into the polymer matrix, the crystallinity of the polymer decreases because the BSG particles interfere with the crystallisation of the polymer. Additionally, with the addition of BSG, the melting peak of the material splits into two peaks together, which occurs because different crystalline structures are formed due to the addition of BSG, which does not allow the crystallisation of the polymer to be uniform.

Figure 6 shows different thermal transitions: on the one hand, the glass transition (T_g_) of PLA is observed at a temperature between 58 and 62 °C and a peak between 131 and 132 °C, which corresponds to the melting temperature of PLA. In addition, two thermal transitions are observed. A glass transition temperature at −40 °C and another melting temperature at 120 °C. These transitions correspond, as shown in other works [35], to the glass transition and melting temperature of PBAT. This is consistent with the information in the technical data sheet provided by the supplier, where the material is described as “a biodegradable compound and additives”. In the DSC curves, it is observed that, as in the BioPE formulations, the glass transition and melt temperatures decrease with the addition of bagasse, with the melting peaks being smaller with the addition of BSG, due to less crystallisation of the polymer with the addition of BSG.

Figure 7 shows a peak corresponding to PP melting that can be seen between 163 and 165 °C. This temperature decreases as the BSG content increases, as is the case with BioPE and PLA formulations.

The TGA curves of the BioPE, PLA, and PP formulations are shown in Figure 8, Figure 9 and Figure 10, where the loss of weight with the temperature is shown on the left, and, on the other hand, the derivative of the loss of weight with respect to the temperature is shown on the right. 

Figure 8 shows a first drop in weight that corresponds to the degradation of the BioPE, and when it contains BSG, a second drop is observed when the O_2_ gas enters (600 °C), due to the degradation of the oxidisable groups of the BSG. It is also observed that the degradation of the polymer begins at lower temperatures when BSG is added and is between 264 °C and 452 °C. This is due to earlier degradation of BSG. 

The maximum degradation of BioPE is observed by the first derivative (right graph), and it can be seen that the maximum degradation is at slightly higher temperatures with the addition of BSG. The maximum degradation temperatures of BioPE formulations are between 478 °C and 491 °C.

Figure 9 shows a first drop in weight that corresponds to the degradation of PLA and a second drop when O_2_ gas enters from the degradation of the oxidisable groups of PLA and BSG. It is also observed that the degradation of the polymer begins at lower temperatures when BSG is added, and the degradation initiation temperature is between 261 °C and 347 °C. This is due to earlier degradation of BSG. Maximum degradation is observed by the first derivative (right graph), and it can be seen that the maximum degradation of PLA is at slightly higher temperatures with the addition of BSG. The maximum degradation temperatures of PLA formulations are between 368 °C and 398 °C.

Figure 10 shows a first drop in weight that corresponds to the degradation of PP, and when it contains BSG, a second drop is observed when the O_2_ gas enters due to the degradation of the oxidisable groups of the BSG. It is also observed that polymer degradation begins at lower temperatures when BSG is added. In PP formulations, the degradation temperature is between 259 °C and 366 °C. This is due to earlier degradation of BSG. Maximum degradation is observed by the first derivative (right graph), and it can be seen that the maximum degradation of PP is at slightly higher temperatures with the addition of BSG. The maximum degradation temperatures of PP formulations are between 424 °C and 438 °C.

Table 4 shows a summary of the glass transition temperatures (T_g_) for BioPE, PLA, and PP formulations.

The MFI results obtained for the different BGS biocomposites are shown in Table 5.

In order to compare the results, they are shown in Figure 11 as a bar graph. A decrease in material fluidity is observed with increasing bagasse fibre content for both BioPE and PLA biocomposites. As expected, this increase in the viscosity is due to the incorporation of the fibre into the material that hinders its flow because the fibres are in a solid state during the test. That is, fibres restrict the mobility of the polymeric matrix macromolecules when subjected to the tensile stress. As there is a lower proportion of molten material, the material flows less. For PP biocomposites, however, this trend is not observed in the PP composites, presenting similar values (or even slightly higher) with the incorporation of BGS. This behaviour could be attributed to the higher extrusion-compounding temperatures required for preparing the PP composites, as shown in Table 2, reaching 220 and 225 °C, which could degrade the fibres in some extent, leading to a small increase in the flow of the material. In general, processing temperatures of cellulosic fibres are limited to 200 °C or a higher for short periods [40].

Table 6 shows the hardness Shore D and tensile test results: Young’s modulus (E), maximum stress (σ_m_), and elongation at maximum stress (ε_m_).

It can be observed that for BioPE biocomposites, Young’s modulus (E) increases with the BGS addition from 592 to 608 MPa, and this value increases up to 680 MPa in the case of BioPE with 30 wt% BGS. This represents an increase of about 15% of Young’s modulus (E), indicating that the material is more rigid due to the fibre content, also considering the intrinsic flexibility of the BioPE. As indicated before, fibres can restrict the mobility of the polymeric chains during the tensile test, and this behaviour trend is in agreement with the MFI results obtained. Therefore, the maximum stress (σ_m_) is reduced with the addition of beer bagasse from 23.0 to 19.9 MPa (BioPE70BSG30), which means a decrease in the maximum stress (σ_m_) of 14%. Due to this increase in rigidity with the addition of bagasse, the elongation at maximum stress (ε_m_) is reduced from 6.5% to 3.5% with the addition of 30 wt% BGS. This increase in Young’s modulus (E) and decrease in maximum strength (σ_m_) and then the BGS fibre addition to the BioPE produce a reduction in the ductile behaviour of the material. This behaviour can be observed in the stress–strain curves shown in Figure 12.

The same behaviour is observed in the PP biocomposites. The BGS fibre causes a significant variation in the Young’s modulus, reaching an increase of about 97% in the composite with 30 wt% BGS. In comparison, the maximum stress (σ_m_), however, undergoes small variations, with values ranging from 20.2 MPa to 22.8 MPa, and the elongation at this maximum stress also decreases with the fibre content due to the increase in the rigidity of the material and brittleness, decreasing the ductility of the material. This behaviour can be observed in the stress–strain curves shown in Figure 13.

The lower ductility of the BGS-filled BioPE and PP is reflected in the impact resistance of the biocomposites, as a progressive decrease in impact strength is observed (Figure 4). This decrease represents an 80% and 55% in the BioPE and PP biocomposites, respectively.

In contrast to BioPE and PP, the developed PLA biocomposites present an improvement in ductility with the BGS fibre compared to the more rigid unfilled PLA (see Young’s modulus, E_t_, and elongation at maximum stress, **ε_mt_**, of PLA in Table 6). Although the fibres can restring the mobility of the polymer molecules, as suggested by the decrease obtained in MFI, in this case, the decrease in E**_t_** value with BGS addition can be attributed to the intrinsic structure of PLA.

Compared to the hydrocarbon and apolar chemical structure of PE or PP, which have a certain flexibility due to the Van der Waals intermolecular forces, PLA contains carbonyl groups that form intermolecular or intramolecular hydrogen bonds with the hydrogen of the adjacent molecular chain [41], causing rigidity in the material, and the crystalline PLA chains have weak mobility. For this reason, by introducing the fibres into the polymeric matrix, the formation of all the possible hydrogen bonds is prevented to a certain extent and the rigidity decreases, so that the Et obtained from the composites is lower than that of unfilled PLA. Once the BGS is introduced, the modulus increases slightly with the fibre content following the same trend observed in PE and PP composites.

E value of the PLA composites increases with the BGS content, from 10 to 30 wt%, becoming slightly more rigid, as expected with the filler content and in the same way as the BioPE and PP composites. The maximum stress (σ_m_) is reduced with the addition of bagasse from 23.2 MPa for PLA90BSG10 then to 18.6 MPa for 20 and 30 wt%. The elongation at this maximum stress (ε_m_) does not vary much with the addition of bagasse. This behaviour can be observed in the stress–strain curves shows in Figure 14.

In this case, however, although the BGS biocomposites are not so rigid, a small decrease in impact strength (up to 32%) is produced. 

The drop in impact resistance (Figure 15) indicates that increasing the fibre content produces greater brittleness of the material due to the lower homogeneity of the material that transmits the impact energy less efficiently through the bulk material before the break of the specimen.

Regarding hardness, a variation is observed when adding 10 wt% of BGS to the polymers, slightly increasing the BioPE and PP biocomposites and decreasing the PLA-based composites. As the BGS content is increased, the trend is also the same for BioPE and PP composites, in that it slightly increases with BGS content up to 30 wt%, mainly due to the increase in the rigidity of the material with the BSG, which act as a filler. Hardness barely decreases in the PLA composites. This is probably due to the compatibiliser contribution to ductility, which compensates the increase in rigidity produced by the BSG incorporation in PLA.

Figure 16, Figure 17 and Figure 18 show SEM micrographs of the fractures obtained from the Charpy impact test at 100× (left) and 500× (right) of BioPE, PLA, and PP biocomposites, respectively.

For BioPE biocomposites (Figure 16), a heterogeneity is clearly observed in the three developed formulations, with a highly laminated structure and large gaps (marked with circles and red arrows) caused by the beer bagasse (BSG) fibres and a certain lack of union of these with the polymeric matrix. This lack of binding between the matrix and the fibre is clearly observed in the SEM image (f), in which a large fibre has been captured (green circle) and a gap of relevant size (≃20−30 μm) is visible (red arrows). In addition, high roughness is observed in the fibre with certain holes inside, which are partially covered by polymeric material. These heterogeneities are the cause of the drop in mechanical properties observed in the previous sections, such as the reduction in maximum stress (σ_m_), elongation (ε_m_), and the reduction in impact strength.

In PLA biocomposites (Figure 17), there is a clear heterogeneity in three formulations, with holes (red circles and arrows) and BSG split by the impact (green arrows in (d) and (f)). In this case, a greater adhesion of the fibres to the polymeric matrix is observed than in mixtures of BioPE with BSG. However, in some of the fibres, there is a gap between the fibre and the matrix (red arrows and circles). This distance is estimated between 6.5 μm and 10 μm, being a distance less than the distance observed in the micrographs of the BioPE blends with BSG. Despite the greater interaction between the bagasse and the polymeric matrix, it should be noted that heterogeneities are still observed in the blends, which are what cause the fall of the impact strength as well as the maximum stress drop in tensile test.

In PP biocomposites (Figure 18), a heterogeneous material is observed, with irregularities in the polymer matrix, being able to appreciate the presence of BSG and the breakage of some of the fibres after the impact test (circle in (e)). But, unlike BioPE and PLA composites, there is greater integration of the fibre with the matrix. However, as can be seen in the SEM image (n), there are some gaps (red arrows) between the matrix and the fibre since the integration is not complete. This means better behaviour of the compatibilising additive, which explains why the growth of Young’s modulus with the addition of fibre is more pronounced and the drop in impact strength is less pronounced than with BioPE and PLA biocomposites.

## 4. Conclusions

The main conclusions obtained from the results can be summarised as follows:

For BioPE biocomposites, an increase in the rigidity of the material with the increase in BSG content can be observed. This increase in rigidity is reflected with the increase in the tensile modulus, decrease in maximum strain, drop in impact resistance, and increase in the hardness of the material. In addition, there is a lack of compatibility of the matrix and bagasse fibre despite the use of a compatible additive. This behaviour is observed in the micrographs obtained with SEM, in which gaps between the fibre and the polymer matrix have been observed. From the addition of BSG, the BioPE degrades before, a decomposition occurs in two–three phases, and the temperature of initiation of degradation should be considered as a limiting factor in the processing of these materials. 

For PLA biocomposites, the following effects can be observed: a decrease in the stiffness of the material with increasing BSG content. This increase in stiffness is reflected in the decrease in the tensile modulus, a decrease in maximum stress, and a decrease in the hardness of the material. In addition, there is higher compatibility than BioPE biocomposites with bagasse; although, there is still some lack of compatibility, as shown by SEM analyses, where gaps between the matrix and the fibre are observed. However, these gaps are smaller, and the fibre is better distributed throughout the sample. Furthermore, there is a decrease in the glass transition temperature, melting temperature, and an increase in the maximum degradation temperature. Finally, there is a decrease in MFI with an increase in BSG content by not melting the fibre and slightly impeding the flow of the material.

For PP biocomposites, an increase in the rigidity of the material with the increase in BSG content can be observed. This increase in rigidity is reflected in an increase in the tensile and modulus, a decrease in elongation in maximum strain, a drop in impact resistance, and an increase in hardness. Additionally, no significant changes in melting temperature are observed, as is the case with mixtures of BioPE and PLA biocomposites. Lastly, for changes in MFI, there is an increase with the addition of 10% bagasse, but then MFI decreases with increasing the content by 20 wt% and 30 wt% bagasse. This increase may be due to an onset of bagasse degradation.

Overall, it is observed that the biocomposite with the highest rigidity is BioPE70BSG30, that is, the composite of BioPE with 30 wt% of BSG. On the other hand, the biocomposite with the lowest rigidity is PLA70BSG30, a PLA biocomposite with a BSG content of 30 wt%. These conclusions can be a starting point to plan different potential industrial applications of these materials as garden furniture, kitchenware, or consumer goods.

## Figures and Tables

**Figure 1 polymers-16-02916-f001:**
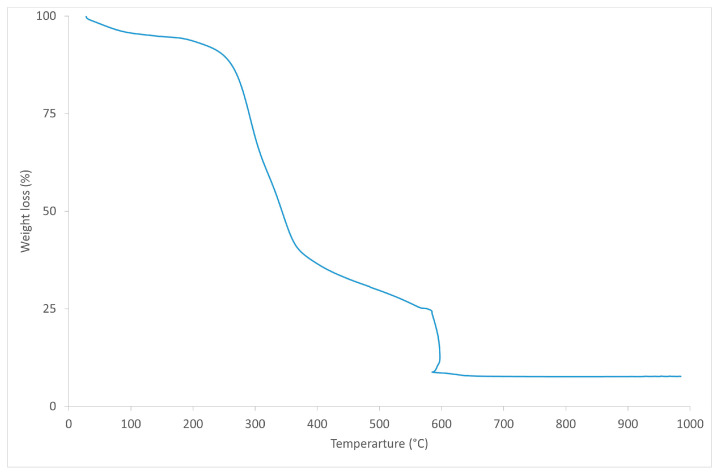
TGA curve of BSG.

**Figure 2 polymers-16-02916-f002:**
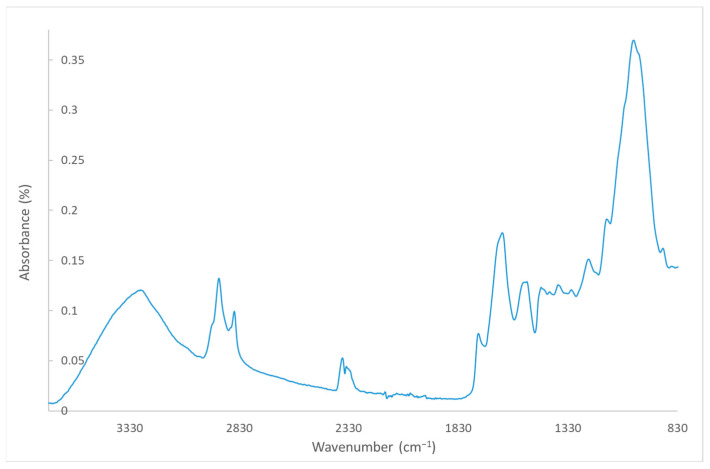
FTIR profile for BSG.

**Figure 3 polymers-16-02916-f003:**
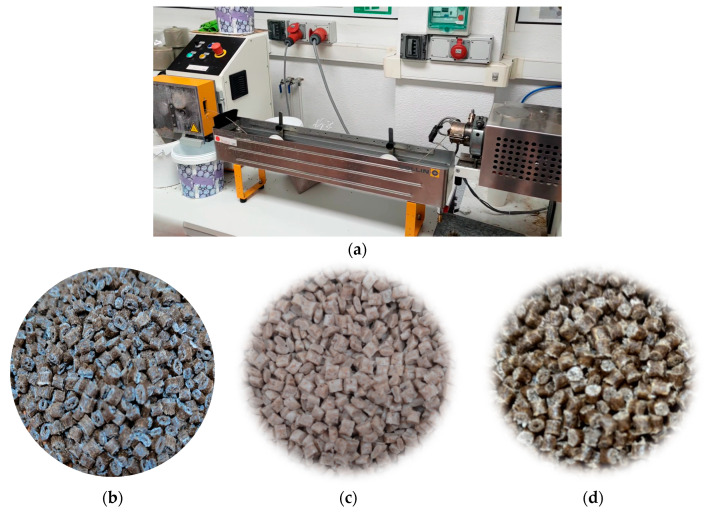
(**a**) Extrusion compounding equipment; (**b**) BioPE biocomposites; (**c**) PLA biocomposites; and (**d**) PP biocomposites.

**Figure 4 polymers-16-02916-f004:**
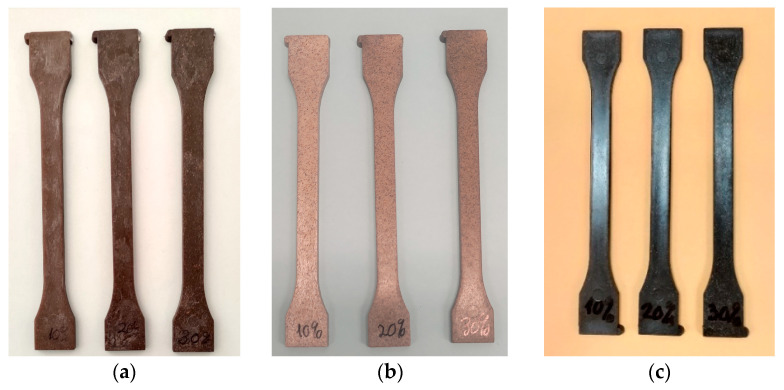
Injected samples of (**a**) BioPE biocomposites; (**b**) PLA biocomposites; and (**c**) PP biocomposites.

**Figure 5 polymers-16-02916-f005:**
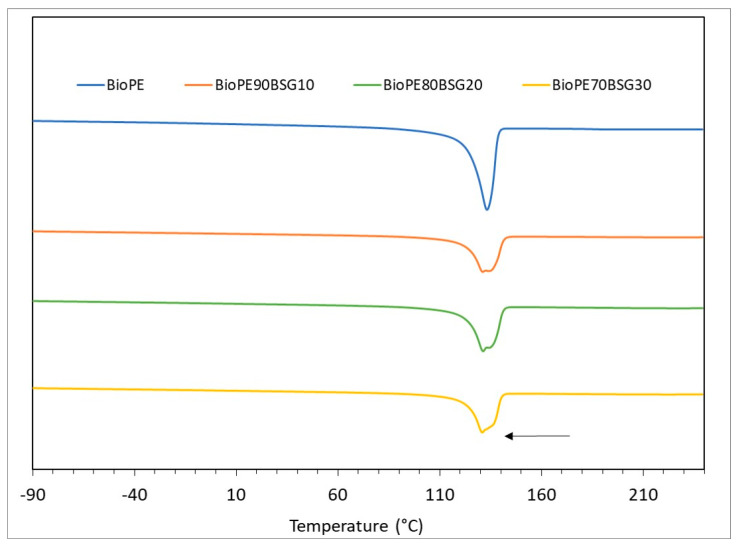
DSC curves of BioPE biocomposites.

**Figure 6 polymers-16-02916-f006:**
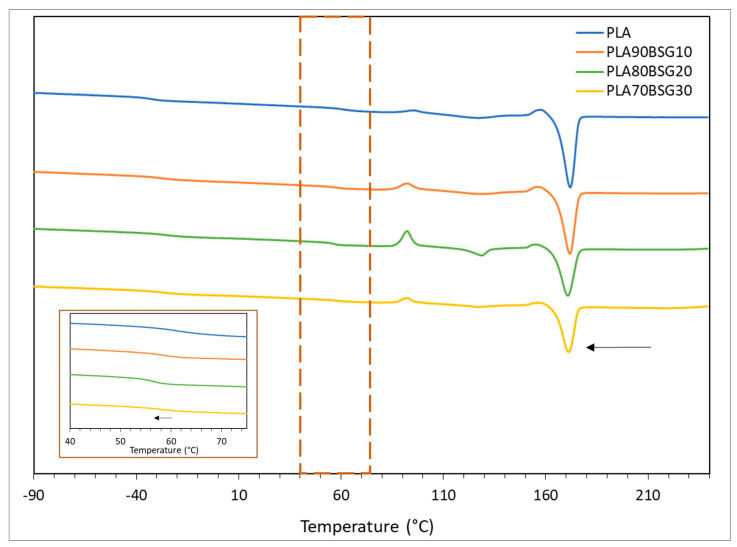
DSC curves of PLA biocomposites.

**Figure 7 polymers-16-02916-f007:**
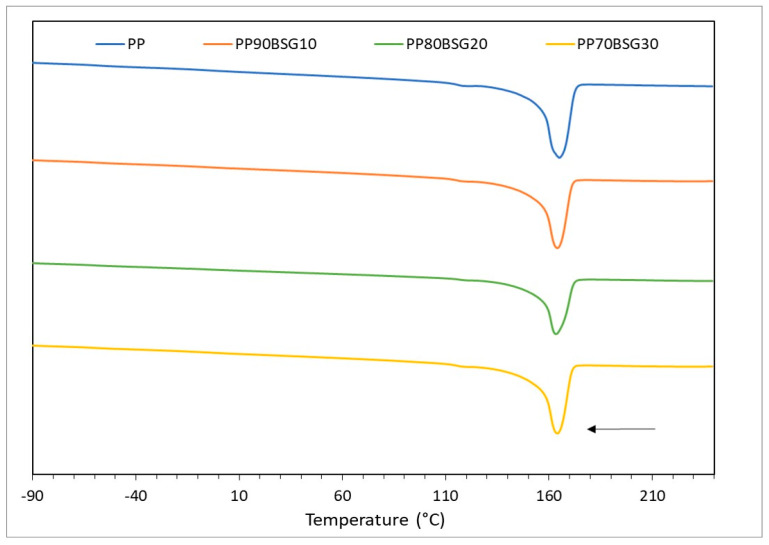
DSC curves of PP biocomposites.

**Figure 8 polymers-16-02916-f008:**
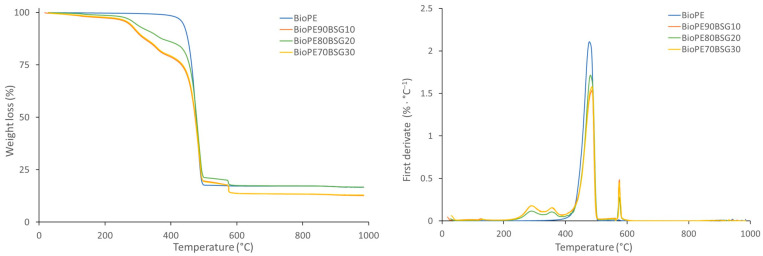
TGA curves of BioPE biocomposites.

**Figure 9 polymers-16-02916-f009:**
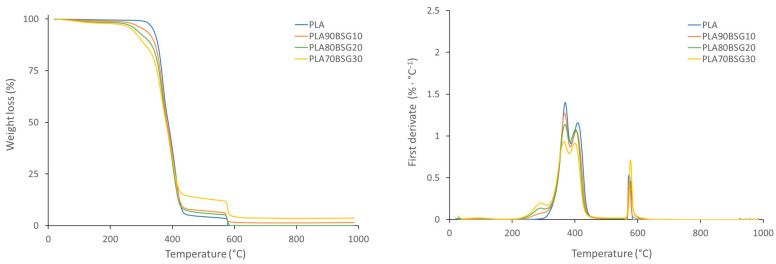
TGA curves of PLA biocomposites.

**Figure 10 polymers-16-02916-f010:**
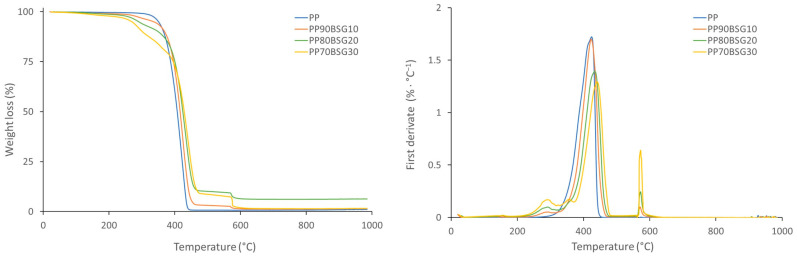
TGA curves of PP biocomposites.

**Figure 11 polymers-16-02916-f011:**
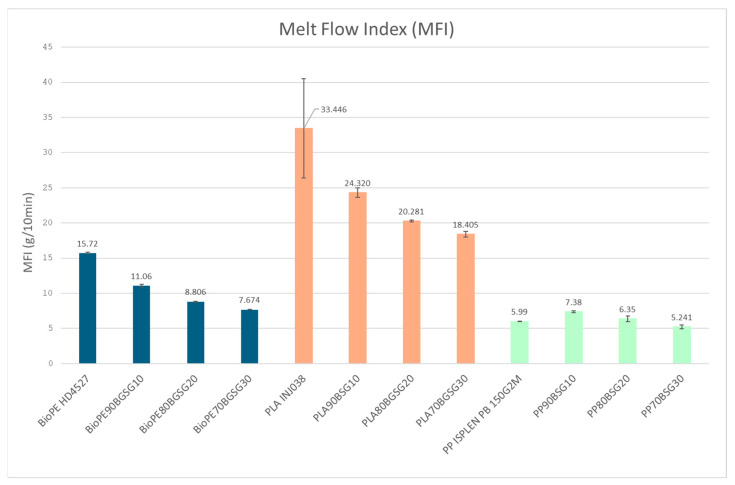
Bar graph MFI results.

**Figure 12 polymers-16-02916-f012:**
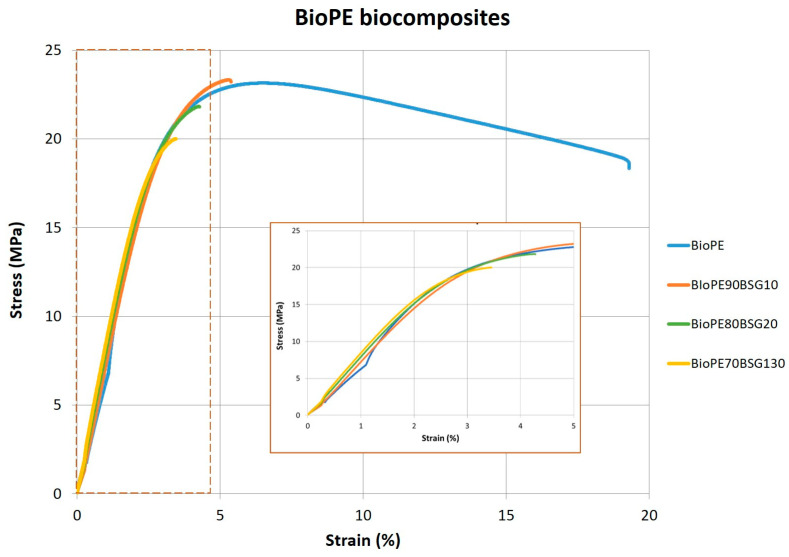
Stress–strain curves for BioPE biocomposites.

**Figure 13 polymers-16-02916-f013:**
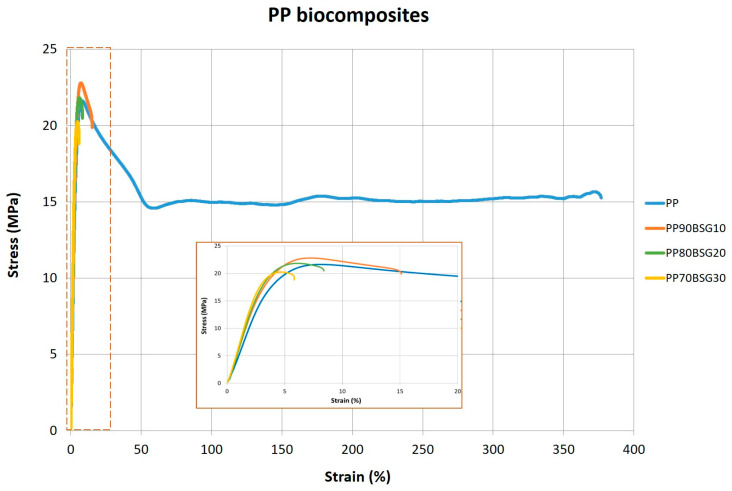
Stress–strain curves for PP biocomposites.

**Figure 14 polymers-16-02916-f014:**
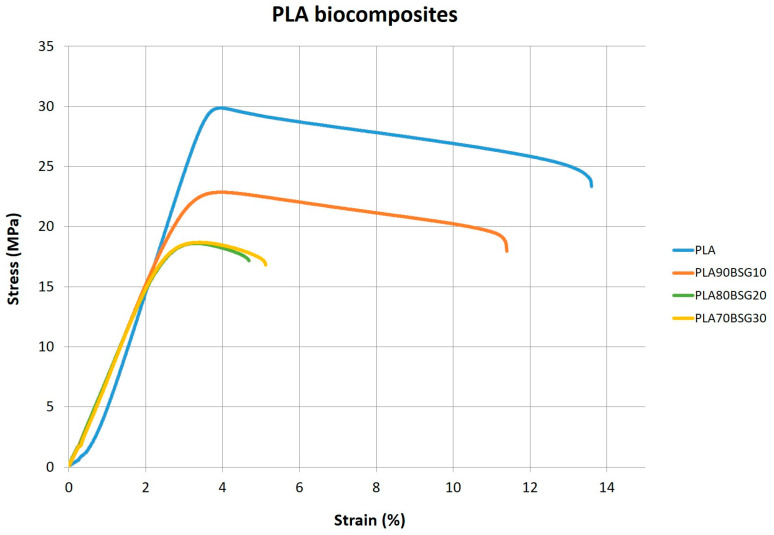
Stress–strain curves for PLA biocomposites.

**Figure 15 polymers-16-02916-f015:**
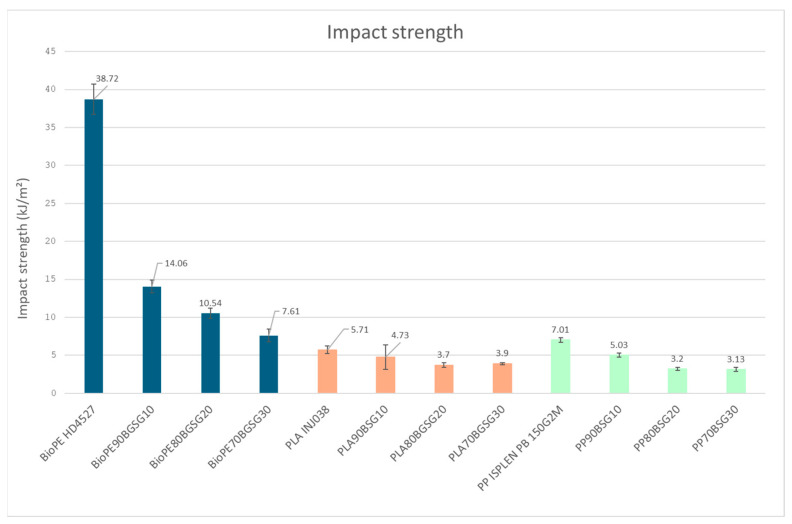
Bar graph impact strength results.

**Figure 16 polymers-16-02916-f016:**
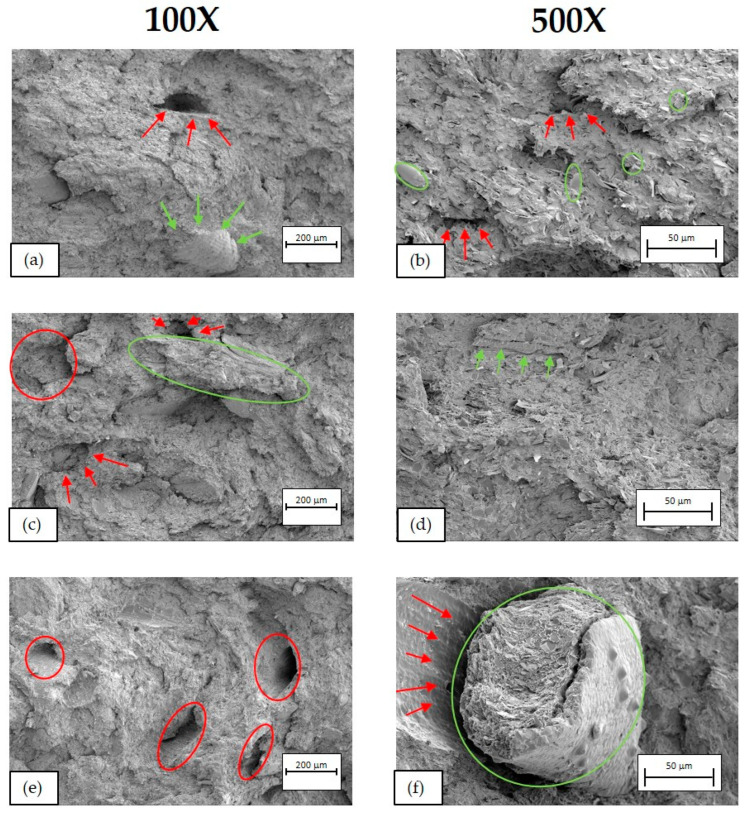
SEM micrographs of BioPE biocomposites. (**a**) BioPE90BSG10 at 100×, (**b**) BioPE90BSG10 at 500×, (**c**) BioPE80BSG20 at 100×, (**d**) BioPE80BSG20 at 500×, (**e**) BioPE70BSG30 at 100× and (**f**) BioPE70BSG30 at 500×.

**Figure 17 polymers-16-02916-f017:**
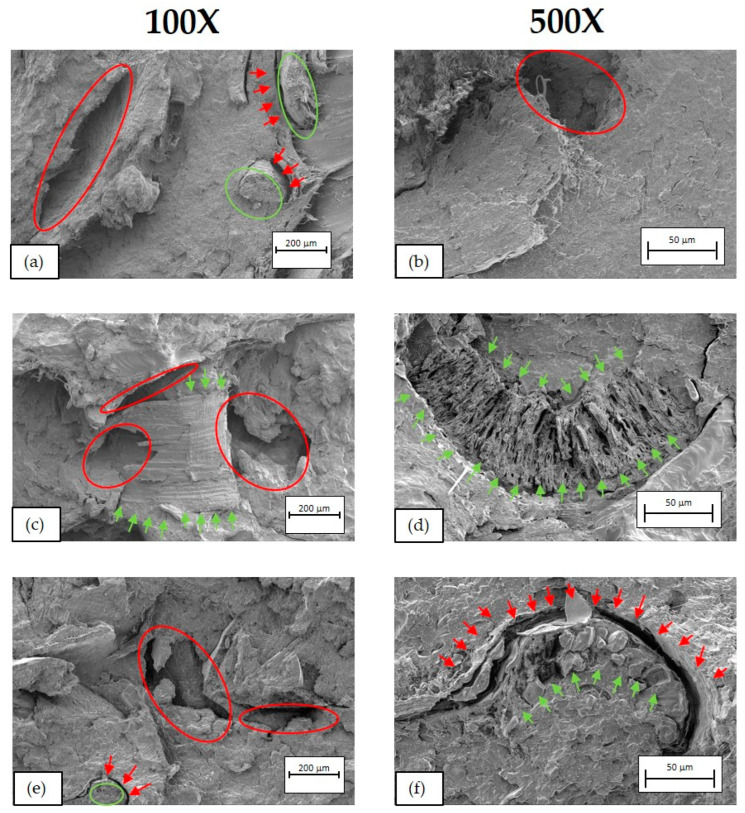
SEM micrographs of PLA biocomposites. (**a**) PLA90BSG10 at 100×, (**b**) PLA90BSG10 at 500×, (**c**) PLA80BSG20 at 100×, (**d**) PLA80BSG20 at 500×, (**e**) PLA70BSG30 at 100× and (**f**) PLA70BSG30 at 500×.

**Figure 18 polymers-16-02916-f018:**
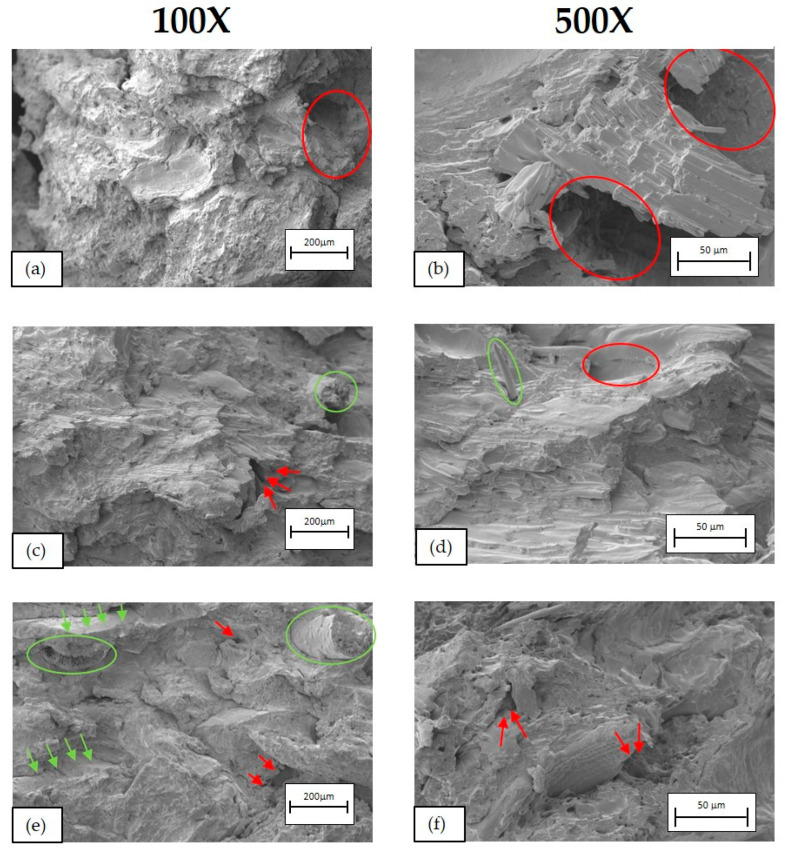
SEM micrographs of PP biocomposites. (**a**) PP90BSG10 at 100×, (**b**) PP90BSG10 at 500×, (**c**) PP80BSG20 at 100×, (**d**) PP80BSG20 at 500×, (**e**) PP70BSG30 at 100× and (**f**) PP70BSG30 at 500×.

**Table 1 polymers-16-02916-t001:** Developed biocomposites formulations.

Biocomposite Name	Polymer Used	Polymer Percentage (wt%)	BSG Percentage (wt%)	Compatibilising Additive	Additive Quantity (phr)
BioPE90BSG10	BioPE	90	10	POLYBOND 3009	3
BioPE80BSG20		80	20		
BioPE70BSG30		70	30		
PLA90BSG10	PLA	90	10	XIBOND 250	
PLA80BSG20		80	20		
PLA70BSG30		70	30		
PP90BSG10	PP	90	10	POLYBOND 3200	
PP80BSG20		80	20		
PP70BSG30		70	30		

**Table 2 polymers-16-02916-t002:** Extrusion compounding conditions.

	Temperatures (°C)						Speed (rpm)
	Zone 1 (°C)	Zone 2 (°C)	Zone 3 (°C)	Zone 4 (°C)	Zone 5 (°C)	Zone 6 (°C)	
BioPE biocomposites	40	75	185	195	190	190	75
PLA biocomposites	40	65	180	185	190	190	65
PP biocomposites	50	65	185	205	220	225	65

**Table 3 polymers-16-02916-t003:** Injection moulding conditions for the processing of test specimens.

Name	Temperature Profile (°C)	Injection Speed (rpm)	Compaction Time (s)	Compaction Pressure (bar)	Cool-down Time (s)	Loading Speed (s)
BioPE biocomposites	200-200-190-180-35	60	10	400	40	70
PLA biocomposites	190-190-180-170-35	60	20	350	50	70
PP biocomposites	195-195-185-180-35	70	10	300	40	80

**Table 4 polymers-16-02916-t004:** Results of thermal characterisation. Glass transition temperatures (Tg) are collected in a summarized form for BioPE, PLA, and PP formulations.

	Materials	T_g_ (°C)	T_m_ (°C)	T_onset_ (°C)	T_dmax_ (°C)
BioPE biocomposites	BioPE HD4527	-	132.94 ± 0.31	451.82 ± 1.65	478.36 ± 0.57
	BioPE90BSG10	-	131.25 ± 0.14	263.81 ± 0.63	491.6 ± 4.68
	BioPE80BSG20	-	131.39 ± 0.04	262.56 ± 0.30	480.60 ± 0.23
	BioPE70BSG30	-	131.09 ± 0.14	264.60 ± 0.53	486.14 ± 0.98
PLA biocomposites	PLA INJ038	62.08 ± 0.61	171.99 ± 0.21	346.56 ± 2.11	368.90 ± 0.27
	PLA90BSG10	57.22 ± 0.67	171.71 ± 0.04	340.70 ± 1.99	367.34 ± 0.06
	PLA80BSG20	57.50 ± 1.65	170.79 ±0.13	338.33 ± 1.35	368.35 ± 0.53
	PLA70BSG30	58.33 ± 0.56	171.10 ± 0.05	261.2 ± 0.75	398.17 ± 4.61
PP biocomposites	PP ISPLEN PB 150G2M	-	165.31 ± 0.63	366.19 ± 3.41	423.86 ± 3.00
	PP90BSG10	-	164.3 ± 0.24	379.57 ± 1.16	431.94 ± 5.82
	PP80BSG20	-	163.58 ± 0.14	260.16 ± 3.25	433.41 ± 2.31
	PP70BSG30	-	163.59 ± 0.09	259.54 ± 0.12	437.65 ± 4.08

**Table 5 polymers-16-02916-t005:** Results of melt flow index of biocomposites.

	Materials	MFI (g/10 min)
BioPE biocomposites	BioPE HD4527	15.7 ± 0.108
	BioPE90BSG10	11.1 ± 0.188
	BioPE80BSG20	8.81 ± 0.069
	BioPE70BSG30	7.67 ± 0.036
PLA biocomposites	PLA INJ038	33.4 ± 7.04
	PLA90BSG10	24.3 ± 0.490
	PLA80BSG20	20.2 ± 0.13
	PLA70BSG30	18.4 ± 0.42
PP biocomposites	PP ISPLEN PB 150G2M	5.99 ± 0.03
	PP90BSG10	7.38 ± 0.14
	PP80BSG20	6.35 ± 0.40
	PP70BSG30	5.24 ± 0.26

**Table 6 polymers-16-02916-t006:** Tensile test and hardness results.

		Tensile Test Results			Hardness Shore D
		E_t_ (MPa)	σ_m_ (MPa)	ε_m_ (%)	
BioPE biocomposites	BioPE HD4527	592 ± 48.4	23.0 ± 0.212	6.5 ± 0.047	62 ± 2
	BioPE90BSG10	608 ± 87.1	24.1 ± 0.429	4.9 ± 0.29	76 ± 1
	BioPE80BSG20	630 ± 30.6	21.7 ± 0.406	4.2 ± 0.20	74 ± 1
	BioPE70BSG30	680 ± 17.4	19.9 ± 0.238	3.5 ± 0.068	73 ± 1
PLA biocomposites	PLA INJ038	825 ± 97,4	30.4 ± 0.607	3.3 ± 0.13	96 ± 1
	PLA90BSG10	621 ± 30.8	23.2 ± 0.0961	4.0 ± 0.037	79 ± 1
	PLA80BSG20	644 ± 6.39	18.6 ± 0.423	3.3 ± 0.061	78 ± 0
	PLA70BSG30	635 ± 9.49	18.6 ± 0.467	3.4 ± 0.051	78 ± 1
PP biocomposites	PP ISPLEN PB 150G2M	343 ± 27.9	21.4 ± 0.434	8.1 ± 0.062	75 ± 1
	PP90BSG10	614 ± 6.78	22.8 ± 0.206	7.2 ± 0.041	78 ± 1
	PP80BSG20	636 ± 7.69	21.8 ± 0.157	6.0 ± 0.036	80 ± 1
	PP70BSG30	677 ± 8.01	20.2 ± 0.213	4.7 ± 0.041	82 ± 1

## Data Availability

The original contributions presented in the study are included in the article material, and further inquiries can be directed to the corresponding author.

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
