# Peer review of "Study of Mechanical and Thermal Properties of Environmentally Friendly Composites from Beer Bagasse"

_polymers, 2024, doi:10.3390/polym16202916_

Round 1

Reviewer 1 Report

Comments and Suggestions for Authors

Attached

Comments on the Quality of English Language

Minor English correction needed

Author Response

  1. Line 24: Too many key words are there. Remove the excess. Remove microscopy after SEM.

Keywords referring to characterization techniques have been eliminated in the text

  1. Line 29: what the plastic mean by the author? Is thermoplastic is not plastic?

The word “plastic” has been changed to “thermoplastic" in the document because, as you tell me, this term is more correct in the scientific field, and more specifically in materials science.

  1. Line 72: can use only BSG because it has been used earlier.

This proposed change has been made to the text by removing "beer bagasse fibres".

  1. Line 87: Author deals with environmental friendly, circular economy and sustainability. Is polyethylene biodegradable? Why do the author chose it?

As indicated in line 96 of the introduction to the manuscript, the PE used is "a polyethylene from renewable sources", this means that the monomers used for the synthesis of PE do not come from oil but are obtained from resources of renewable origin. A biobased polymer, a biobased and biodegradable polymer and a conventional polymer from a non-renewable source are chosen to do a complete study in different polymers.

  1. Line 107: the author chose various additives for each matrix polymers. In introduction the importance of the additives was not discussed. How does the author chose additives. Include the references.

To solve this interesting comment, the authors have added a paragraph at the end of the introduction where the type of compatible additives as well as their function are explained in detail (lines 98-103). In addition, section 2.1 expands with more detailed information on the characteristics and functionality of the three additives used in this study (Line 135 – line 165).

  1. How does the author decided to go for the 3Phr as additive quantity irrespective of the different matrices?

This proportion of compatibilizing additive is chosen because it is an intermediate value within the range (1phr – 5phr) commonly used in different works where additives with a compatibilizing effect are used.  In addition, as it was intended to add the largest amount of BSG, a value that is not very small is chosen, to be able to make the amount of 30% BSG compatible). Some jobs where they use between 1 phr and 5 phr are:

Quiles-Carrillo, L., Balart, R., Boronat, T. et al. Development of Compatibilized Polyamide 1010/Coconut Fibers Composites by Reactive Extrusion with Modified Linseed Oil and Multi-functional Petroleum Derived Compatibilizers. Fibers Polym 22, 728–744 (2021). https://doi.org/10.1007/s12221-021-0024-z

Jordà-Reolid, M.; Moreno, V.; Martínez-Garcia, A.; Covas, J.A.; Gomez-Caturla, J.; Ivorra-Martinez, J.; Quiles-Carrillo, L. Incorporation of Argan Shell Flour in a Biobased Polypropylene Matrix for the Development of High Environmentally Friendly Composites by Injection Molding. Polymers 2023, 15, 2743. https://doi.org/10.3390/polym15122743

Jorda-Reolid, M.; Gomez-Caturla, J.; Ivorra-Martinez, J.; Stefani, P.M.; Rojas-Lema, S.; Quiles-Carrillo, L. Upgrading Argan Shell Wastes in Wood Plastic Composites with Biobased Polyethylene Matrix and Different Compatibilizers. Polymers 2021, 13, 922. https://doi.org/10.3390/polym13060922

Jorda, M.; Montava-Jorda, S.; Balart, R.; Lascano, D.; Montanes, N.; Quiles-Carrillo, L. Functionalization of Partially Bio-Based Poly(Ethylene Terephthalate) by Blending with Fully Bio-Based Poly(Amide) 10,10 and a Glycidyl Methacrylate-Based Compatibilizer. Polymers 2019, 11, 1331. https://doi.org/10.3390/polym11081331

  1. How many repetitions were made on each test?

This has varied depending on each trial carried out. For the thermal tests, only one repetition was performed.  But, after reading this commentary, the authors have considered it very appropriate to increase the number of repetitions to three, to obtain statistical data on mean and deviation. That is why the data shown in Table 4 have changed the results to a mean and deviation made with three repetitions.

For the tensile and hardness Shore D tests, 5 repetitions are carried out, for the impact and MFI tests 10 repetitions. In section 2.3 (2.3.1 - 2.3.3) of the manuscript, the number of repetitions performed in each trial has been added.

  1. Include the graphs for the respective tests in the thermal discussion.

The graphs are included in section 3  (Figure 5 – Figure 10) of the manuscript. Table 5 has also been commented on as a summary of the results obtained from the thermal characterization.

  1. It is expected to discuss the reasons for the Tg, Tm, Tonset, Tdmax of each composite preparation.

The thermal incantations have been commented on in detail as you suggest, making reference to the results shown in the graphs of Figure 5 – Figure 10 added in response to the previous comment

  1. In fig 3 include the decimal points instead of comma.

It has been changed in the manuscript. (Now Figure 11 and Figure 115. Figures 5 to 10 have also used periods instead of commas).

  1. In table 5 check PP biocomposite specification. BioPE90BSG10.

We change in the manuscript (Table 5) “BioPE...” for “PP...”).

  1. Discuss the reason for the decrease in flow of PP composite.

This explanation has been added in the paragraph below Table 5 (line 314 – line 320 and 354-358), comparing the behavior of PP with bioPE and PLA.

  1. Table 6. Check the notation provided in the content with the table representation. Eg. elongation at maximum stress (εm), εσmt.

In Table 6, "εσmt" is changed to "εm”, “smt" is changed to “sm”.

  1. MFI value is reducing for the PE and PLA composites. Whereas the Young’s modulus increased for the PE composites and decreased for the PLA composites. Discuss in detail.

The authors have added information that explains this phenomenon in detail in the text between table 6 and figure 12 (line 326 and 339) and later between figure 12 and figure 13 (line 354 – 376). At this point, the structures of each polymer and how the addition of BSG can affect the mobility of polymer chains have been related.

  1. In PP composite the tensile values of virgin PP and 10% fibre are 21.4 ± 0.434 and 22.8 ± 0.206. The corresponding elongation values are 8.1 ± 0.062 and 7.2 ± 0.041. there is no much deviation in the values. But the Young’s modulus values are 343 ± 27.9 and 614 ± 6.78. Almost doubled. Provide the stress strain diagram for these results and justify.

Stress-strain diagrams have been added to the manuscript (Figure 12, Figure 13 and Figure 14). Figure 14 shows this diagram for PP biocomposites. In this case, when representing the four biocomposites, it is observed that the elongation at break of PP is much higher than the rest of the formulations with BSG. This demonstrates the fragility that BSG brings to this material. But this phenomenon does not allow us to observe the effect of the module when BSG is added to the PP. That is why a graph has been represented where the results are shown at the beginning of the test, so that the part of the module is magnified. In this zoom of the diagram, it did look like a major increase in the module.

Reviewer 2 Report

Comments and Suggestions for Authors

Comments on the Quality of English Language

Upper-case and lower-case letters are sometimes misleading. 

The tense of the whole paper needs to be checked. The authors sometimes used past tense, sometimes used present tense!!!

Author Response

  1. On page 5, line 163, O2 flow should be O2 flow. Some letters in the capture of Figure 4 should be the lower case. The authors need to more careful and serious when writing the manuscript.

We have changed this sentence from O2 to O2. This change is on the current line 214.

2.For Table 5, some MFIs have 3 significant figures, some have 4 figures, and some have 5 figures. How did the author determine the significant figures? Have the authors calculated the uncertainty? Please make sure the values have the same significant figures.

At this point, the specific standard for the test has been revised: ISO 1133, which indicates that the results must be expressed with 3 significant figures and no more than 2 decimal places. The changes are made in Table 5

3.The authors put all SEM images together, which is very hard for audience to read. Please consider having three separate Figures for three composites. Please also have arrows showing the morphology change when increasing the beer bagasse concentration.

The SEM images have been placed in three figures (Figure 16, Figure 17 and Figure 18). Pointing out with circles and arrows the interactions between the matrices and the BSG.

4.The SEM images show heterogeneities in fiber-matrix interactions. While the authors discuss this briefly, a more in-depth analysis of how these structural defects affect mechanical properties such as impact resistance could add value to the study.

The authors have added red arrows and circles to the microscopy figures (Figure 16, Figure 17 and Figure 18) to mark the gaps between the polymer matrix and the fiber and the BSG fibers with green circles and arrows.

5.The characterization of used beer bagasse was missing, such as FTIR, TGA, etc.

TGA y FTIR curves are added in lines 120-134 and these graphics are commented (Figure 1 and Figure 2)

6.The TGA and DSC graphs were missing. For materials science, it is necessary to include them.

The authors have included these graphs, and they have been commented on together with the tabulated results of Table 5.

7.The Discussion part was weak. It seems the authors were only reporting the results, no comparison to the literature, no deep analysis on what happened and why. Please read some similar work and add some necessary literature for discussion.

The discussion part of the results has been expanded by the authors.

8.The authors chose 10 wt%, 20 wt%, and 30 wt% of beer bagasse, but why were these specific concentrations selected?

In the work shown in the manuscript, we wanted to study the effect of adding BSG to different matrices, in order to increase the renewable character of the polymers and reduce the amount of polymer. To this end, it was decided to carry out a study increasing the percentage of BSG from 10% to 10%, from raw material to 30% BSG, since at this percentage the compound began to be more brittle, and it was not possible to process the material with the equipment available in our processing laboratory.

9.A more precise explanation of the intended industrial applications would provide better context for the study’s importance.

This explanation has been added in the fifth paragraph of the introduction (line 85 – line 93). Some applications in which BSG formulations could be used are generically reflected. Although some of these formulations with percentages of 10% and 20% BSG have been studied at an industrial level, obtaining promising results. The authors do not have the consent of the companies that have collaborated in carrying out these tests, to show their results. For this reason, it is not shown in greater detail.

10.The novelty of using beer bagasse as a filler is interesting. However, the paper could benefit from a more detailed comparison with other bio-filler-based composites to highlight what makes beer bagasse particularly suitable or superior in performance or cost.

In the introduction to the present manuscript, the authors explained the high quantities of beer produced at European level. These values could justify the economic and environmental benefits of using waste with high levels of production, due to the high consumption and production of beer worldwide. However, it is not possible to make an exhaustive comparison of the economic and technical benefits of using this fibre in thermoplastic polymers in this manuscript because it would require a very dedicated study of economic and technical data of different fibres and flours of natural origin. However, the authors find this point very interesting and we value making this comparison in a future work, reflecting it in the form of a review.

11.For scientific paper, the conclusion should be written as paragraphs instead of using bullets, which making the scientific article more like a report.

We have corrected the conclusions section using cohesive paragraphs and adding the appropriate connectors to give coherence to the text.

Round 2

Reviewer 2 Report

Comments and Suggestions for Authors

The size of the x-axis and y-axis and their names in Figures 5-10 are too small. Please consider making them larger. 

Author Response

The size of the x-axis and y-axis and their names in Figures 5-10 are too small. Please consider making them larger.

The font size of the axes as well as the axis titles and legend of these graphs have been increased for better visualization.